# OpenReview forum: "Personalization Under Value Conflict: Resolving Contradictory Preferences with Paired Fine-Tuning"
_ICLR.cc/2026/Conference — ICLR 2026 Conference Withdrawn Submission_

### Official Review · Reviewer_HNWA · 2025-10-28

**Soundness:** 3
**Presentation:** 2
**Contribution:** 2
**Rating:** 4
**Confidence:** 3

**Summary:**

Humans often adjust their preferences when facing different situations, which causes models trained on a single preference to underperform. In this work, the authors propose **Preference-Paired Fine-Tuning (PFT)**, which optimizes the models to align contradictory preferences under the same scenario simultaneously. They present a dataset called **Value Conflict Dilemma (VCD)** that consists of contradictory paired examples, and apply supervised fine-tuning on it. The experimental results indicate that PFT performs better than all single-preference baselines for both positive and negative preferences.

**Strengths:**

- The motivation for aligning the models for both contradictory preferences at the same time is well-defined.
- The inclusion of human studies is appreciated, as it strengthens the evaluation.

**Weaknesses:**

**About the method**

- In Section 3.5, the authors mention that the PFT-trained model can quickly adapt to an ICL scenario, but do not explain why. The training process itself does not appear to include any ICL-related design.

**About the experiments**

- The ICL experiments in Section 4.3 are not clearly explained. For example, the source of the user history data is unclear, and it is not specified which model was used. Without these details, the results are difficult to interpret.
- In line 418, the authors claim that single-preference trained models cannot generalize well to other preference types without giving the supporting evidence.

**About the writing**

- The use of the synchronous update method for the main approach is mentioned only in the captions of Tables 1 and 2. This should be clearly stated in the main text to distinguish it from the baselines.
- Human evaluation is mentioned in line 387 but is only described in the appendix, making it hard to follow.
- Figure 5 appears to contain duplicated bars.

**Questions:**

- Does the improvement mainly come from using contradictory preferences, or simply from sampling multiple preferences within the same scenario? Have the authors tested a dataset built with independent preferences and fine-tuned the models on it?
- For the single-preference baselines, do they use the same number of training examples (1000), or does the main method actually use 1000 *pairs* of examples?
- The negative preference results seem consistently lower than the positive ones. Since both types of preferences should be neutral with respect to human values, what might explain this difference?
- This framework appears adaptable to RLHF-based methods such as PPO or GRPO. Have the authors explored or considered experiments in that direction?

---

### Official Review · Reviewer_8vgH · 2025-11-01

**Soundness:** 2
**Presentation:** 2
**Contribution:** 1
**Rating:** 2
**Confidence:** 3

**Summary:**

The paper studies personalization when a single user can toggle between contradictory preferences. It introduces Preference-Paired Fine-Tuning (PFT): train with both sides of each pair via either alternating updates or a summed loss, then steer at inference with ~3 in-context examples. The paper also releases a synthetic Value Conflict Dilemma (VCD) dataset and evaluates on VCD and selected tasks.

**Strengths:**

1. The paper presents a clear problem framing of value conflict alignment and a simple recipe that plugs into SFT/DPO pipelines.

2. VCD dataset could be potentially useful as it defines explicit contradictory pairs and is human-checked for label quality.

**Weaknesses:**

1. **Baselines for multi-objective control are missing:** Several directly comparable multi-objective alignment/controlled generation methods, e.g., [1,2,3], are not compared. These methods also aim to train one steerable policy that trades off conflicting objectives during inference.

2. **Limited technical contribution:** PFT is essentially cross-entropy on both sides of a pair (either alternated or summed), with standard gradients and weights. The proposed fast personalization path is simply a user context-conditioned generation by adding user history to the prompt, which has already been explored in several inference-time steering methods, e.g., [2,3].

3. **Scalability and robustness:** The paper studies k=2 (binary contradictory), but many user preferences are multi-dimensional and non-exclusive. The method’s effectiveness to scale to >2 dimensions or interactively changing preferences is unknown.

[1] Zhou, Zhanhui, et al. "Beyond One-Preference-Fits-All Alignment: Multi-Objective Direct Preference Optimization." Findings of the Association for Computational Linguistics ACL 2024. 2024.

[2] Wang, Kaiwen, et al. "Conditional Language Policy: A General Framework For Steerable Multi-Objective Finetuning." Findings of the Association for Computational Linguistics: EMNLP 2024. 2024.

[3] Guo, Yiju, et al. "Controllable Preference Optimization: Toward Controllable Multi-Objective Alignment." Proceedings of the 2024 Conference on Empirical Methods in Natural Language Processing. 2024.

**Questions:**

See Weaknesses.

---

### Official Review · Reviewer_Jmge · 2025-11-01

**Soundness:** 2
**Presentation:** 3
**Contribution:** 2
**Rating:** 4
**Confidence:** 3

**Summary:**

The paper proposes Preference-Paired Fine-Tuning (PFT), a method that trains a single LLM on paired demonstrations of contradictory preferences to enable dynamic personalization under value conflict. It introduces the Value Conflict Dilemma (VCD) dataset and shows that PFT outperforms baselines in both classification and open-ended generation tasks.

**Strengths:**

1.	The work tackles the important and underexplored challenge of personalizing LLMs when user preferences conflict.
2.	It introduces VCD, a high-quality, human-validated dataset that supports future research on value conflicts.
3.	The paper is well-written and easy to read.

**Weaknesses:**

1.	Despite introducing the terms “asynchronous” and “synchronous” update strategies, the method is essentially standard supervised fine-tuning on preference-conditioned paired data and offers limited novelty.
2.	The paper models preferences as strict binary opposites, whereas real-world preferences often exist on a spectrum or are contextually blended, limiting the framework’s applicability to nuanced user behaviors.
3.	The evaluation primarily focuses on the single-dimensional VCD benchmark, lacking assessment in multi-dimensional or finer-grained preference settings, which limits the validation of the method’s generalizability.
4.	The paper primarily compares against general alignment methods (e.g., SFT, DPO, CAA) but omits comparisons with recent specialized personalization techniques. This limits the assessment of PFT’s relative advantages in the broader landscape of personalized LLMs.

**Questions:**

See the weaknesses.

---

### Official Review · Reviewer_dRzQ · 2025-11-10

**Soundness:** 2
**Presentation:** 2
**Contribution:** 2
**Rating:** 2
**Confidence:** 4

**Summary:**

This paper tackles the challenge of aligning LLMs with heterogeneous and contradictory user preferences. The paper proposes Preference-Paired Fine-Tuning (PFT), a method that fine-tunes a single model on both sides of a contradictory preference pair, enabling the model to handle opposing preferences without requiring separate models for each preference direction. The paper additionally introduces the Value Conflict Dilemma dataset and show that PFT can combine paired training with lightweight in-context adaptation to better match individual preference histories.

**Strengths:**

- Well-motivated problem. The paper addresses an important limitation in current LLM alignment approaches: most methods optimize for universal preferences rather than handling individual-level preference diversity and conflicts.
- Comprehensive experiments. Tests on multiple model sizes and families (Qwen, LLaMA), multiple baselines (SFT, DPO, CAA), multi-format evaluation (multi-choice classification with “pick-one” and “select-all-that-apply” protocols, and open-ended generation scored by GPT-4o-mini), as well as ablations on dataset size and preference-pair combinations.
- New Dataset. VCD focuses specifically on value-conflict scenarios and includes human validation. Even if synthetic, the attention to contradictory labeling could be useful.

**Weaknesses:**

- Methodological Novelty. The core idea is essentially training on both sides of a preference pair simultaneously. This is a relatively incremental modification to standard SFT. The mathematical formulation (especially the synchronous update in Eq. 5-7) is just standard multi-task learning with weighted losses
- Strength of Claims vs. Results. Some narrative framing seems overstated relative to the reported improvements. For example, several gains in the tables are modest, and certain baselines (e.g., DPO in single-preference directions) outperform PFT in their own setting. The claim that PFT “significantly” improves open-ended generation would benefit from a more tempered interpretation.
- Missing Critical Operational Detail. Several important methodological details are missing or insufficiently described in the main text. For example, regarding multi-choice evaluation, the main text does not explain how model outputs are converted into selected choices, nor how generation is constrained for the “All” setting. Additionally, given that VCD is positioned as one of the principal contributions, the main text provides only high-level construction details.
- Unclear How Explanations are Used or How Important They Are. Although explanations are repeatedly emphasized as part of the single-choice training data (“triplet of <scenario, preference, explanation>”), the paper does not run any experiments isolating the impact of these generated explanations. Their actual contribution remains unclear.

**Questions:**

- What is the empirical impact of including generated explanations during training or inference? Since explanations appear prominently in the data pipeline, an ablation (e.g., training with vs. without explanations) seems necessary to understand their influence.
- How exactly is the user-history context constructed for ICL? Are histories sampled directly from the training distribution, synthesized in a principled way, or drawn from held-out samples?
- Why are there duplicate entries in Figure 5?
- The abstract mentions a ~40% reduction in data requirements compared to single-preference fine-tuning, and the conclusion similarly claims that the method is more “data-efficient than SFT and DPO.” However, none of the reported results in the main text seem to justify these numbers. Could you clarify how this number was computed and how the experiments support this conclusion?

---

### Note · Authors · 2026-01-07

I have read and agree with the venue's withdrawal policy on behalf of myself and my co-authors.